# The folding propensity of α/sulfono-γ-AA peptidic foldamers with both left- and right-handedness

Peng Teng [1,2,4✉], Mengmeng Zheng[1,4], Darrell Cole Cerrato [1], Yan Shi[1], Mi Zhou[1], Songyi Xue[1], Wei Jiang[1,3], Lukasz Wojtas[1], Li-June Ming[1], Yong Hu[3✉] & Jianfeng Cai [1✉]

The discovery and application of new types of helical peptidic foldamers have been an attractive endeavor to enable the development of new materials, catalysts and biological molecules. To maximize their application potential through structure-based design, it is imperative to control their helical handedness based on their molecular scaffold. Herein we first demonstrate the generalizability of the solid-state right-handed helical propensity of the $4_{13}$-helix of L-α/L-sulfono-γ-AA peptides that as short as 11-mer, using the high-resolution X-ray single crystallography. The atomic level folding conformation of the foldamers was also elucidated by 2D NMR and circular dichroism under various conditions. Subsequently, we show that the helical handedness of this class of foldamer is controlled by the chirality of their chiral side chains, as demonstrated by the left-handed $4_{13}$-helix comprising 1:1 D-α/D-sulfono-γ-AA peptide. In addition, a heterochiral coiled-coil-like structure was also revealed for the first time, unambiguously supporting the impact of chirality on their helical handedness. Our findings enable the structure-based design of unique folding biopolymers and materials with the exclusive handedness or the racemic form of the foldamers in the future.

[1] Department of Chemistry, University of South Florida, Tampa, FL, USA. [2] Institute of Drug Discovery and Design, College of Pharmaceutical Sciences, Zhejiang University, Hangzhou, Zhejiang, P. R. China. [3] College of Engineering and Applied Science, Nanjing University, Nanjing, Jiangsu, P. R. China. [4] These authors contributed equally: Peng Teng, Mengmeng Zheng. ✉email: pengteng@zju.edu.cn; hvyong@nju.edu.cn; jianfengcai@usf.edu

For decades, the development of synthetic foldamers[1,2] that mimic the structure and function of natural biopolymers has boosted markedly. These synthetic oligomers are endowed with enhanced resistance toward proteolytic degradation and sequence diversity, along with promise in biomedical and material applications[2]. Meanwhile, akin to natural peptides, they can bind to various biomolecular targets such as proteins[3–5], membranes[6–9], RNAs[10], and so on, with the ability to fold into conformationally stable structures. To date, foldamer backbone including β-peptides[11], peptoids[12], β-peptoids[13], oligoureas[14], aza-peptides[15], α-aminoisobutyric acid (Aib)[16], oligoproline[17], cis-β-aminocyclo-propane carboxylic acids (cis-β-ACCs)[18], hybrid peptides[19], aromatic amide foldamers[20,21], etc., have been well established especially by crystallographic analysis, leading to fruitful applications in molecular self-assembly and recognition[22,23]. However, as the natural features an endless repertoire of structure and function, the identification and creation of new types of foldamer scaffolds is still challenging but yet to be achieved[2]. Gratifyingly, heterogeneous foldamers involved backbones containing subunits from two or even three classes of molecular scaffolds, e.g., α-peptides containing β- or γ-amino acid residues[24,25], have emerged to be a promising strategy to significantly increase the availability of molecular structures and biological functions[19,26]. Despite the limited success of the hybrid foldamers, the development of new classes of hybrid foldamers and the precise control of their helical propensity remain largely unexplored.

Since 2011, γ-AApeptides (stemming from chiral PNA backbone[27]) have emerged as a new class of peptidomimetics with advantages of immense chemical diversity and enhanced resistance toward proteolytic degradation[28]. This class of peptidomimetics has shown robust promise in biomedical and material sciences[29]. However, it was highly challenging to understand the helical folding propensity of this type of oligomers, since they are distinct from the double helix formed by PNA/PNA or PNA/DNA which are stabilized by Watson–Crick base pairing rather than intramolecular hydrogen bonding[30]. Recently, we have reported two types of right-handed helices formed by heterogeneous 2:1 L-α/D-sulfono-γ-AA hybrid oligomers ($4.5_{16-14}$-helix)[31] and heterogeneous 1:1 L-α/L-sulfono-γ-AA hybrid oligomers ($4_{13}$-helix)[32], and one left-handed helix comprising homogeneous L-sulfono-γ-AA oligomers ($4_{14}$-helix)[33], all of which advanced our capability to gain insight of rational design in the future. Among them, 1:1 L-α/L-sulfono-γ-AA hybrid oligomers exhibit a right-handed 13-helix pattern, closely resembling that of α-helix. The exploration of this class of foldamer are expected to lead to profound application in functional materials. However, the structures of 1:1 L-α/L-sulfono-γ-AA hybrid oligomers were limited in only a few sequences on the solid state[32,34,35]. Their solution conformation, the generalizability of their folding propensity, including sequence length, both in solid and solution state, has not been systematically explored. In addition, it is known that D-peptides, as enantiomers of L-peptides, form left-handed α-helix, and thus one could postulate that 1:1 D-α/D-sulfono-γ-AA hybrids are doomed to adopt left-handed 13-helix. However, as the molecular scaffold of α/sulfono-γ-AA peptides is different from α-peptides, which contains 50% of chiral side chains and 50% of achiral sulfonyl side chains, an unambiguous folding structure is needed to support the hypothesis. Indeed, the structural principles between L- and D-peptide partners of such foldamers could only be ideally interpretive should the atomic-resolution structural characterization of such heterochiral assemblies be achieved[36]. However, to date very few X-ray crystal structures of the racemic form derived from a α-helical peptide corresponding to the segment of the protein have been reported[37,38].

Herein we first report our comprehensive investigation on the folding of shorter oligomers by down-sizing the recent $4_{13}$-helix (heterogeneous 1:1 L-α/L-sulfono-γ-AA hybrid foldamers)[32]. The difference in the folding propensity of the exact ratio of 1:1 and 1:1 + α (Fig. 1a) in the solid state were both investigated. In addition, we have also successfully revealed a left-handed helical foldamer based on the unprecedented 1:1 D-α/D-sulfono-γ-AA hybrid. For the first time, the short right-handed 1:1 L-α/L-sulfono-γ-AA hybrid foldamers and the left-handed 1:1 D-α/D-sulfono-γ-AA hybrid foldamers were systematically investigated, which unambiguously strengthens the folding architecture in this class of oligomers under various circumstances. Our results demonstrated that the helical handedness of this class of foldamer is controlled by the chirality of their chiral side chains and irrelevant to achiral sulfonyl side chains. In the last, a racemic structure of 1:1 α/sulfono-γ-AA hybrid was demonstrated by the high-resolution signal crystal X-ray diffraction to show the interaction in the heterochiral coiled-coil. Our results shed light on the structure-based design of unique folding biopolymers and functional materials with individual handedness or the racemic form of the foldamers.

## Results and discussion

**Sequence synthesis.** Based on the reported 15-mer foldamer[32], we synthesized a series of 1:1 L-α/L-sulfono-γ-AA hybrid

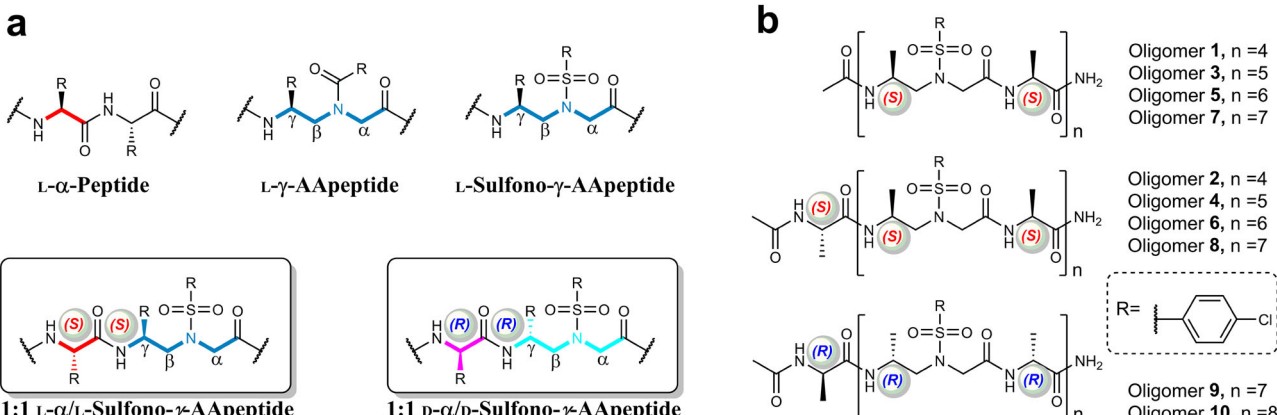

**Fig. 1 Chemical structures of oligomers. a** General structures of L-α-peptides, L-γ-AApeptides, L-sulfono-γ-AApeptides, 1:1 L-α/L-sulfono-γ-AApeptides, and 1:1 D-α/D-sulfono-γ-AApeptides. **b** 1:1 oligomer evaluated in the current study, both the exact ratio of 1:1 (**1**, **3**, **5**, and **7**) and 1: 1 + α (**2**, **4**, **6**, **8**, **9**, and **10**) types of oligomers were included.

oligomers (Supplementary Scheme S1; Figs. S1; S2; Supplementary Table S1) with the shorter length, from 8-mer to 14-mer (oligomer **1**–**7**, Fig. 1b), containing the ratio of 1:1 and 1:1 + α. The L-methyl-sulfono-γ-AA with a chlorobenzene sulfonyl group was selected as before to exclude the potential impact of different side chains on the folding propensity[33]. Oligomer **8**, the 15-mer reported previously[32], was also synthesized to investigate the helical conformation in solution.

**High-resolution crystallographic studies of oligomers 4, 6, and 7.** Out of seven oligomers with incremental length (from 8-mer to 14-mer), we obtained crystals for four oligomers from different solvent systems. The 13-mer **6** crystallized readily from slow evaporation of $CH_2Cl_2/CH_3CN$ (20:80, v/v), which are the same solvents suitable for the crystallization of 15-mer **8** but were proven unsuccessful for other oligomers. After screening various combination of solvent, we were gratified to obtain crystals for 11-mer **4** by slow diffusion of pentane into THF. To our most surprise, the 14-mer **7** bearing exact ratio of 1:1 L-α/L-sulfono-γ-AApeptides was extremely reluctant to crystal and could only crystallize from slow evaporation of chloroform, of which the access took the most challenging effort. Consequently, the crystal structures of **4** (Supplementary Data 1), **6** (Supplementary Data 2), and **7** (Supplementary Data 3 and 4) were successfully solved by single crystal X-ray diffraction analysis with resolutions of 1.1, 1.0, and 1.3 Å, respectively (Supplementary Tables S2–S5). It should be noted that 11-mer (oligomer **4**) is the shortest foldamer comprising such type of backbone and with stable defined conformations in the solid state up to now. The shorter oligomer **2** (9-mer) was able to crystallize from THF/pentane multiple times but the crystals were not suitable for X-ray diffraction analysis (poor resolution about 5.00 Å), thus its structure was not solved.

Despite of the various lengths of those oligomers, their crystals reveal the identically right-handed helical scaffold with even helical pitch of 5.34 Å and radius of 3.05 Å (Table 1, Fig. 2a), same as that of foldamer **8**[32]. The intramolecular 13-hydrogen bindings are also as neat and uniform as the $(i \rightarrow i + 4)$ hydrogen bonding pattern with distance of 1.95–2.11 Å (C = O···HN) in the 15-mer **8**. In addition, the 13-mer **6** was crystallized from the same space group $P4_12_12$ as the 15-mer **8**, however, the shorter oligomer 11-mer **4** was crystallized from $P2_1$ space group, so does the 14-mer **7**. Based on these results, we can see the oligomer that is in exact 1:1 L-α/L-sulfono-γ-AApeptide ratio is extremely challenging to crystallize. The presence of terminal α-amino acid in the 1:1 + α type of oligomers prompts the crystallization of the foldamers, since it contributes to an extra set of head-to-tail intermolecular hydrogen bonding in the 1:1 + α type of oligomers to aid in the packing in the crystal lattice, whereas there is only one set of head-to-tail intermolecular hydrogen bonding exists in the exactly 1:1 type of oligomers (Fig. 2b, c). The infinite four-pedal windmill-shaped columns along peptide axis were formed apparently (Supplementary Discussion), owing to the highly

ordered molecular packing governed by head-to-tail intermolecular hydrogen bonds of N–H···O = C type between N- and C-terminals of helices.

In contrast to the α-helix which bears 3.6 residues/turn, these short foldamers contain exactly four side chains per turn. Their side chains are almost perpendicular to the helical axis and pointing away from the axis. Similar to those of α-helices, the side

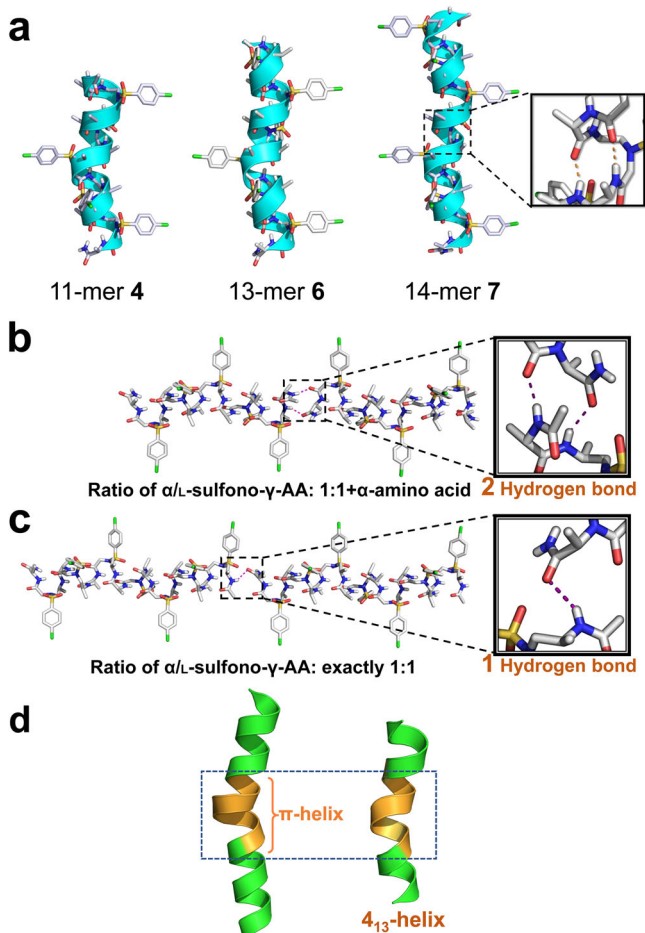

**Fig. 2 Single crystal structures of oligomers 4, 6, and 7. a** Single-crystal structures of $4_{13}$-helix formed by 11-mer **4**, 13-mer **6**, and 14-mer **7**. The intramolecular hydrogen bond was indicated as orange dashed line in the inset. The nonpolar hydrogens were omitted for clarity. Solvent molecules were also excluded from the crystal lattice. **b** Two sets of head-to-tail intermolecular hydrogen bonding in the 1:1 + α type of oligomers; **c** One set of head-to-tail intermolecular hydrogen bonding in the exactly 1:1 ratio type of oligomers. **d** A comparison of π-helix and $4_{13}$-helix. Left: a short seven residue π-helix (orange) embedded within a longer α-helix (green), taken from PDB code 3QHB. Right: a short seven residue fragment of π-helix mimetic (orange), taken from 11-mer **4**.

**Table 1 Parameters of helical structures found in proteins and foldamers consisting of sulfono-γ-AA peptide hybrids.**

| Secondary structure | Backbone | Handedness | Helical pitch $p$ (Å) | Radius of helix $r$ (Å) |
|---|---|---|---|---|
| α-helix | L-α-peptide | Right-handed | 5.4 | 2.3 |
| $3_{10}$ helix | L-α-peptide | Right-handed | 6.0 | 1.9 |
| π-helix | L-α-peptide | Right-handed | 5.0 | 2.8 |
| $4.5_{16-14}$ helix[31] | 2:1 L-α/D-sulfono-γ-AApeptide | Right-handed | 5.1 | 2.6 |
| $4_{14}$ helix[33] | L-sulfono-γ-AApeptide | Left-handed | 5.1 | 2.8 |
| $4_{13}$ helix | 1:1 L-α/L-sulfono-γ-AApeptide | Right-handed | 5.3 | 3.0 |
| D-$4_{13}$ helix | 1:1 D-α/D-sulfono-γ-AApeptide | Left-handed | 5.3 | 3.0 |

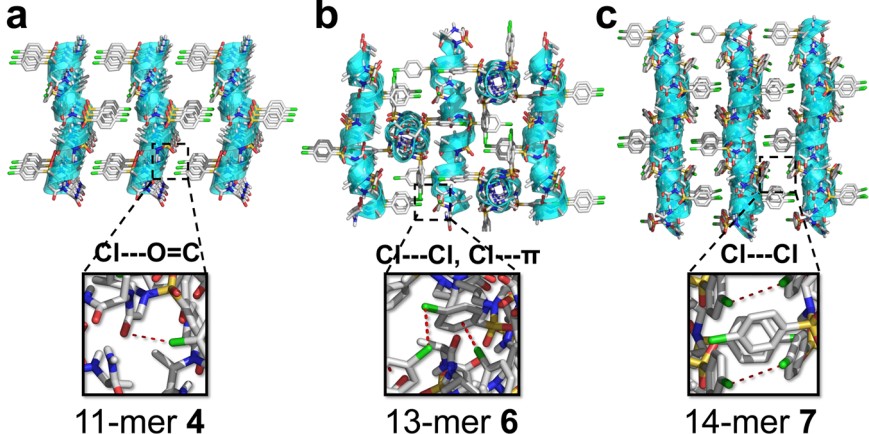

**Fig. 3 Crystal packing of foldamers 4, 6, and 7. a** Crystal packing of 11-mer **4**. **b** Crystal packing of 13-mer **6**. **c** Crystal packing of 14-mer **7**. The intermolecular Cl⋯O=C, C–Cl⋯Cl–C, and C–Cl⋯π interactions are indicated as red dashed line in insets. The disordered solvent THFs/pentanes (in 11-mer **4**), acetonitriles (in 13-mer **6**) or chloroforms (in 14-mer **7**) are excluded from the crystal lattice.

**Table 2 Typical torsion angles (°) in foldamers 4, 6, 7, and 8 based on single crystals.**

|  |  | $\phi$ | $\theta$ | $\eta$ | $\xi$ | $\psi$ | $\phi'$ | $\psi'$ |
|---|---|---|---|---|---|---|---|---|
| Right-handed | **4** | −125.4 | 72.3 | 78.9 | −129.3 | −3.4 | −74.5 | −18.5 |
| | **6** | −117.2 | 64.4 | 72.1 | −140.3 | 10.5 | −72.0 | −24.5 |
| | **7** | −125.5 | 60.9 | 90.4 | −130.2 | 0.6 | −74.4 | −28.1 |
| | **8** | −118.7 | 61.7 | 75.6 | −140.8 | 8.7 | −73.1 | −24.7 |

chains point toward the N-terminus in the $4_{13}$-helix. Further comparison of helical pitch and radius of the $4_{13}$-helix revealed that this type of helices could be potential mimetics of π-helix (Fig. 2d). This is significant since π-helix only exists as very short fragments in α-helix due to its destabilization[39] on the secondary structures, while this class of foldamers is capable of forming much longer π-helix-like structures, they may lead to a new class of foldamer for π-helix mimicry.

In the direction perpendicular to the peptide axis, there exist weak interactions including C-Cl⋯O=C (3.2 Å, in 11-mer **4**), C-Cl⋯π (3.2 Å, in 11-mer **6**), C-Cl⋯Cl-C (3.4 Å, in 11-mer **6** and 14-mer **7**) and other Van der Waals interactions between helices (Fig. 3; Supplementary Fig. S3). The weak interactions and helical shape of oligomers with side chlorobenzene groups drive the arrangement of adjacent helices to be parallel (**4**), antiparallel (**7**), or perpendicular (**6**), all reflected in space group symmetry. Overall, the presence of hydrogen bonds between terminal sides, shape of helices and other weak interactions seem to have a primary impact on crystal packing, particularly along with the intramolecular hydrogen bonding in this class of oligomers, which may enable the design of novel functional materials even with short peptidic length.

Moreover, the mean backbone torsion angles are unambiguously revealed to be quite similar across all structures (Table 2). The torsion angles of α-Ala backbone with $-73 \pm 2°$ and $-24 \pm 4°$ for $\phi'$ and $\psi'$ are different from those of neither α-Ala unit in the $4.5_{16-14}$ foldamers ($-62 \pm 3°$, $-39 \pm 7°$)[31] nor canonical α-helices

($-64 \pm 7°$, $-41 \pm 7°$). The torsion angles of L-sulfono-γ-AA residues with $-122 \pm 5°$ and $65 \pm 7°$ for $\phi$ and $\theta$ are close to that of homogeneous L-sulfono-γ-AA backbone ($-138 \pm 2°$, $66 \pm 5°$)[33], however, the backbone torsion angle $\eta$ ($79 \pm 11°$), $\xi$ ($-135 \pm 6°$), and $\psi$ ($4 \pm 7°$) are sharply distinct from those of homogeneous L-sulfono-γ-AA unit in the left-handed $4_{14}$-helix ($\eta, \xi, \psi = -120 \pm 5°$, $92 \pm 5°$, $-141 \pm 5°$)[33], which is still beyond our understanding at this point, since the homogeneous L-sulfono-g-AA foldamers form unexpected left-handed helices, rather than the right-handed helices in this study. It is not surprising that the torsion angles of L-sulfono-γ-AA residues differ from those of β-sheet or other synthetic peptide scaffold[40–42]. This shows that the average backbone torsion angles of L-sulfono-γ-AA are globally conserved over all the structures, indicating that the folding propensity of this type of foldamers is highly unanimous and predictable. Along with the clear arrangement of the side chain, these parameters will permit the creation of either helical bundles or other defined materials, as well as the rational design of helical structure targeting membrane receptors or protein–protein interactions.

**NMR studies of oligomer 8.** To investigate the atomic-scale details of intramolecular interactions in solution, we conducted the 2D NMR of oligomer **8** at a concentration of 4 mM in $CD_3OH$ at 10 °C. Residue-specific assignments were achieved upon a combination of COSY, gDQFCOSY, zTOCSY, and NOESY spectra (Supplementary Figs. S4, S5 and S6).

Although the NMR spectra of the oligomer exhibited significant signal overlapping, as would be expected, due to the identical side chains through each sulfono-γ-AApeptide building blocks or α-peptide residues, we were able to assign entire protons and identify numerous nuclear Overhauser enhancements (NOEs) correlations, which unambiguously support the defined secondary structure in solution. In addition to strong $i, i + 1$ NOE correlations between NH hydrogens (Fig. 4a), the long-range $i, i + 2$ NOE peaks exist extensively across the foldamer, including $i, i + 2$ NOE correlations between α-peptide NH and methylene/γ-CH protons of L-sulfono-γ-AA two residues away, or between NH and $CH_2/CH_3$ of L-sulfono-γ-AA two residues away in either direction. The chimeric $i, i + 3$ and $i, i + 4$ NOE correlations were also detected among the NH protons of L-sulfono-γ-AA and $CH_3$ three of four residues away. These detected NOE correlations are consistent with $i \rightarrow i + 4$) hydrogen bonding pattern found in crystal structures, indicating predominant defined helical structures of the foldamer in solution.

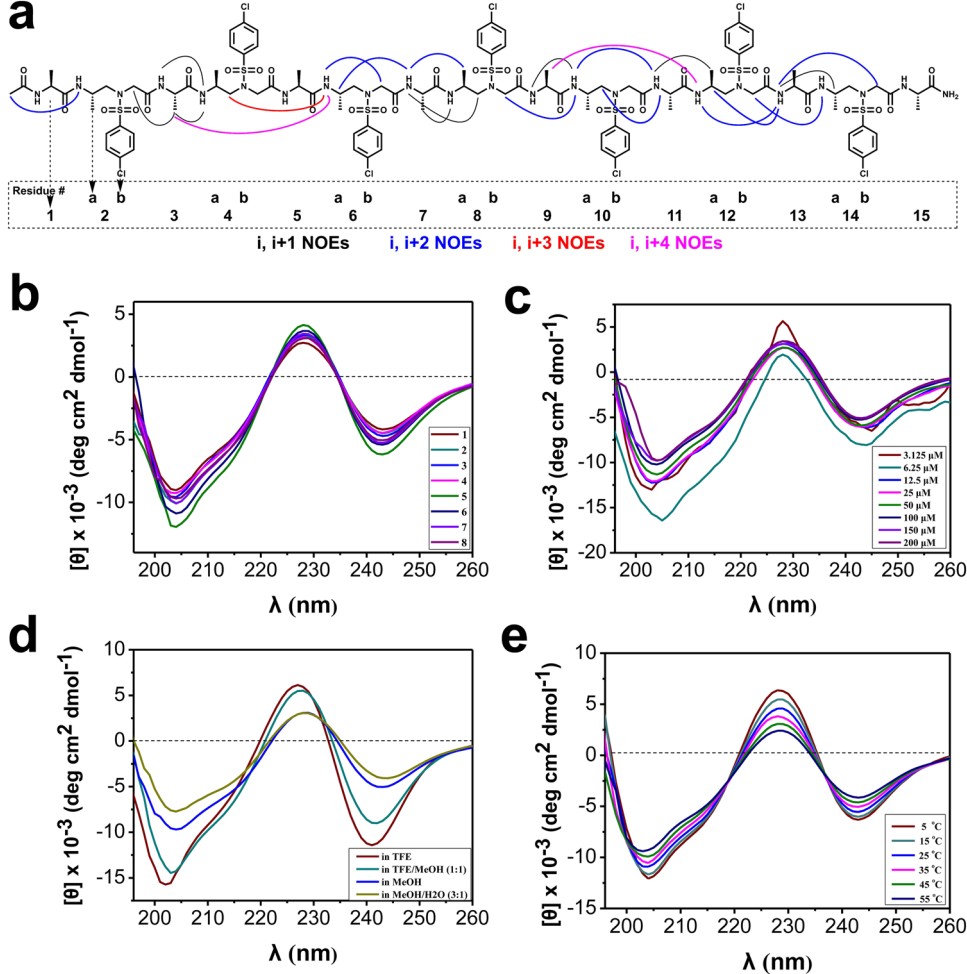

**Fig. 4 Solution structures of foldamers. a** Summary of detected NOESY cross-peaks (200 ms mixing time) of foldamer **8** between protons on nonadjacent residues in CD$_3$OH (4 mM concentration, 10 °C). Four types of NOEs are displayed in different color. Each L-sulfono-γ-AA peptide unit is considered as two residues since it is equal to two α-amino acids in length. **b** CD spectra of compounds **1**–**8** (100 μM) measured at room temperature in CH$_3$OH. **c** CD spectra of compound **8** in CH$_3$OH at various concentrations at room temperature. **d** CD spectra of compound **8** (100 μM) in various solvents at room temperature. **e** CD spectra of compound **8** (100 μM) in CH$_3$OH at various temperatures.

**CD studies**. We next conducted the circular dichroism (CD) spectroscopy of each sequence to correlate their helical structures to the CD spectra. All the eight oligomers displayed similar CD signatures in methanol, although quite different from α-helix. There is a pronounced minimum at 204 nm, similar to the behavior of a helical α/β/γ-peptide[19], and minimum around 242 nm, meanwhile, a maximum at 228 nm (Fig. 4b). The CD trend was also in good agreement with that of the heterogeneous 1:1 L-α/L-sulfono-γ-AA oligomers bearing other miscellaneous side chains[43]. These results indicate that the oligomers have similar solution structures in solution regardless of their lengths and side chains, suggesting robust folding propensity in this class of peptidomimetics.

Concentration-dependence of oligomer **8**, as well as the solvent effect on the helical stability, was also investigated. It is notable that the sequence adopts well-defined helix over the concentration ranging from 3.125 to 200 μM (Fig. 4c). Moreover, it is not surprising that the sequences adopt the best helical conformations in pure trifluoroethanol (TFE) as TFE is a well-known solvent stabilizing the secondary structure, however, in the presence of water the sequence retained an intriguingly good degree of helicity, although the population is somewhat less than that in methanol (Fig. 4d). Lastly, the helically thermal stability of the sequence was also evaluated by temperature-dependent CD

studies. The CD spectra of **8** show no change in shape and only a slight decrease in the CD signal intensity of the minimum at 204 nm and the maximum at 228 nm over the temperature range 5–55 °C (Fig. 4e), indicating high stability of helical sequence of this type in solution. In addition, the curves share isodichroic points at 218 and 237 nm (Fig. 4e), suggesting transitions between two structural states (unfolded and folded) with few intermediate species, the two-state equilibrium also indicated the possible presence of aggregation of the foldamers in solution[44,45].

**Left-handed foldamer**. The conventional α-helices in the proteins conspicuously prefer right-handed configuration[46], while the left-handed peptidic helices were very rare in nature[47], although few examples on the unnatural β-peptides[48] or Aib[9,49] and a few nonpeptidic backbones based helical polymers[50] showed dynamic folding propensity. In sharp contrast, D-amino acid is uncommon in live organisms, and the contribution of D-amino acid on the helical conformation was much less investigated but remains great interest, because the helices consisting of D-amino acid resist degradation by natural proteases[38]. We have systematically demonstrated that the 1:1 α-L/L-sulfono-γ-AA hybrid adopts the right-handed helix vide supra, but it would be attractive to have the atomic level structural information of the

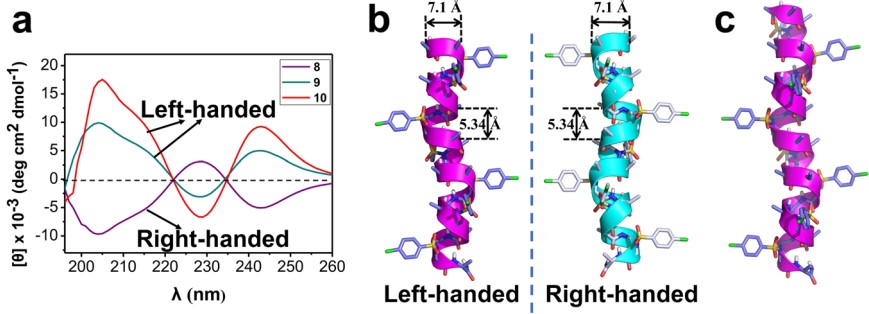

**Fig. 5 Left-handed foldamer. a** CD spectra of 1:1 D-α/D-sulfono-γ-AA hybrids **9** and **10** (100 μM) measured at room temperature in CH$_3$OH, foldamer **8** was included as a mirror comparison. **b** Single-crystal structures of left-handed helix formed by D-15-mer **9**. Single-crystal structures of right-handed helix formed by L-15-mer **8** was employed as a mirror comparison. **c** Single-crystal structures of left-handed helix formed by D-17-mer **10**. The nonpolar hydrogens were omitted for clarity. Solvent molecules were also excluded from the crystal lattice.

**Table 3 Typical torsion angles (°) in foldamers 9 and 10 based on single crystals.**

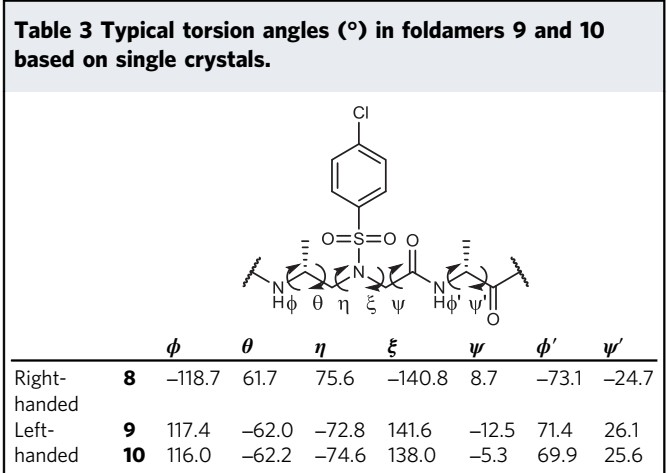

| | | $\phi$ | $\theta$ | $\eta$ | $\xi$ | $\psi$ | $\phi'$ | $\psi'$ |
|---|---|---|---|---|---|---|---|---|
| Right-handed | **8** | −118.7 | 61.7 | 75.6 | −140.8 | 8.7 | −73.1 | −24.7 |
| Left-handed | **9** | 117.4 | −62.0 | −72.8 | 141.6 | −12.5 | 71.4 | 26.1 |
| | **10** | 116.0 | −62.2 | −74.6 | 138.0 | −5.3 | 69.9 | 25.6 |

counterpart of the right-handed helix, namely the left-handed foldamer based on the 1:1 D-α/D-sulfono-γ-AA hybrid (Fig. 1b).

To this end, two oligomers D-15-mer **9** and D-17-mer **10** bearing seven or eight D-sulfono-γ-AA (while bearing eight or nine D-α-amino acid) units were synthesized and characterized by CD to examine their secondary structures. As shown in Fig. 5a, oligomers **9** and **10** displayed mirror-imaged CD spectra of right-handed oligomer **8** in methanol solution, suggesting that the 1:1 D-α/D-sulfono-γ-AA hybrid did form left-handed foldamers in solution, which demonstrated the critical importance of chiral configuration in terms of affording the absolute handedness of this class of foldamers. In the following experiment, both oligomers **9** and **10** readily crystallized from CH$_2$Cl$_2$/CH$_3$CN with decent resolution (1.1 and 1.0 Å, respectively, Supplementary Tables S6, S7), which allowed us to determine the atomically structural information through the X-ray crystallography. As shown in Fig. 5b and c, oligomers **9** (Supplementary Data 5) and **10** (Supplementary Data 6) formed left-handed helical structures with even helical pitch of 5.34 Å and radius of 3.05 Å, same as that found in the right-handed helixes. The same neat intramolecular 13-hydrogen bindings were shown in the left-handed helical structures. The side chains point toward the N-terminus in the left-handed 4$_{13}$-helix, which is the same as that of the right-handed 4$_{13}$-helix since they are in mirror configurations. The space group P4$_3$2$_1$2 in the reasonably left-handed helixes differs from P4$_1$2$_1$2 in the right-handed helixes. In addition, the backbone torsion angles are opposite to that of the right-handed helixes (Table 3). This data unambiguously revealed that the 1:1 D-α/D-sulfono-γ-AA hybrid could form left-handed helixes which are just the mirror

counterparts of the right-handed helixes comprised of 1:1 L-α/L-sulfono-γ-AA hybrid. Such control of handedness of helicity enables more precise and versatile design of these foldamers.

**The heterochiral coiled-coil-like foldamers.** Interactions between the right-handed helices with opposite absolute configurations have been a source of interest for a long time[51,52], notably, mirror-image phage display has emerging as a powerful method to explore D-peptides as protein inhibitors[51,53]. However, the structural principles between L- and D-peptide partners remains underappreciated because of the relatively small number of atomic-resolution structural characterization. A few X-ray crystal structures of the racemic form derived from natural proteins have been reported to reveal the structural principles for the associations of L- and D-polypeptides[54–56], however, the structural information of unnatural-peptides-based heterochiral associations has not been examined.

Our efforts on the racemic recrystallization of foldamers **8** and **9** afforded crystals with X-ray crystal structures of 0.96 Å resolution (Fig. 6a, b). The racemate **11** was crystallized from CH$_2$Cl$_2$/CH$_3$CN using slow vaporization over 2 days. Distinct from the oligomers with single handedness, the racemate **11** (Supplementary Data 7) crystallized in space group I4$_1$/a (Supplementary Table S8). In the crystal structure, left-handed helices contact with the right-handed counterparts closer in space than the helices with single handedness (Fig. 6b), akin to two gears of opposite sense, which further manifests the impact of chiral sides chains on the helical handedness of α/sulfono-γ-AA peptides. This result is similar to the groundwork by Crick who has noticed that the helices in a heterochiral coiled-coil dimer "mesh together" in a manner akin to two gears of opposite sense[57]. In the direction of a and b axis, there are four sets of C＝O⋯H-Ph (a distance of 2.5 Å) and C-H⋯H-Ph (a distance of 2.5 Å) interactions for each helix in between the surrounded helices with opposite handedness (Fig. 6c). In addition, the left-handedness helices form parallel supercoils through the interhelical C-H⋯Cl-Ph interactions (2.7 and 2.9 Å in distance), while in the perpendicular direction, similarly parallel supercoils were formed by the right-handed helices (Fig. 6b, c). The opposite handedness supercoils form weave-like topology in the heterochiral coiled-coil with tighter packing pattern than that in each individual component. As far as we know, the new interaction pattern has not been reported in either natural polypeptides or in the synthetic mimetics[37,38].

## Conclusions
We reported the helical propensity of the 4$_{13}$-helix of α/sulfono-γ-AA peptide hybrid foldamers with length dependency and

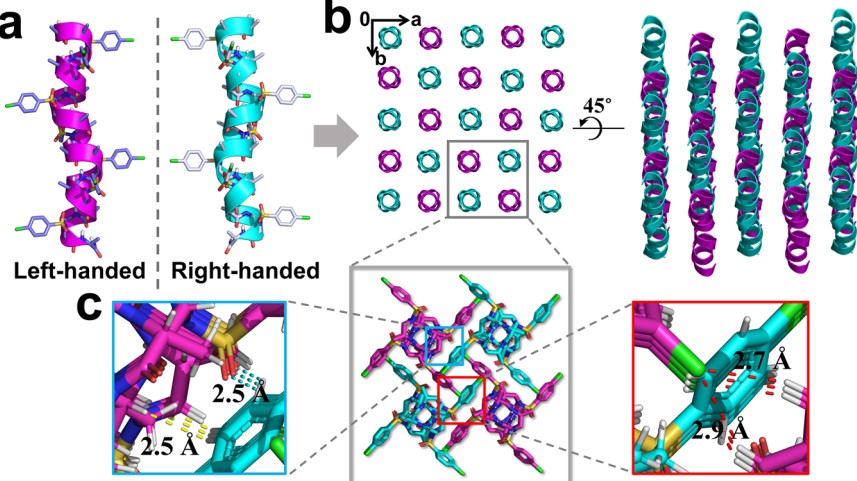

**Fig. 6 Crystal structure of the heterochiral coiled-coil-like 11. a** Single-crystal structures of left-handed helix D-15-mer **9** and right-handed helix L-15-mer **8**. **b** The crystal structures and crystal packing of heterochiral coiled-coil formed by racemic 15-mer **11**. **c** Representative intermolecular interactions in the racemic crystals of **11**. The intermolecular C=O···H–Ph and C–H···H–Ph interactions are indicated as cyan and yellow dashed line, respectively. The interhelical C–H···Cl–Ph interactions are shown as red dashed line. Solvent molecules were excluded from the crystal lattice.

handedness dependency. Well-defined four-petal windmill-shaped helical conformations were strengthened in a series of foldamers with high-resolution structures in both solution and solid state. The solid-state helical structure of 1:1 L-α/L-sulfono-γ-AA oligomer as short as 11-mer can be acquired by crystallographic analysis. Meanwhile, 2D NMR and CD spectroscopic data in organic solvents further support the solution structures, which are indicative of rosy folding in accordance with the atomic level high-resolution X-ray crystal structures. Moreover, the left-handed 1:1 D-α/D-sulfono-γ-AA hybrid foldamers were revealed unambiguously at the atomic level for the first time to prove the seminal proposal. Notably, the racemate of the foldamer forms a heterochiral coiled-coil-like dimer meshing together in a manner akin to two gears of opposite sense. Our findings showed the exquisite control of handedness on this type of heterogeneous backboned peptidic foldamers by chirality manipulation of monomeric building block, which is irrelevant to achiral sulfonyl side chains. The heterochiral coiled-coil dimer also reveals de novo interaction model which represents a starting point for understanding and structural designing associations of tertiary or quaternary assemblies.

## Methods
**General procedure for oligomer synthesis**. All reagents and solvents were purchased from Fisher or Aldrich and used without further purification. Fmoc protected α-amino acids and Rink-amide resin (0.6 mmol/g, 200–400 mesh) were purchased from Chem-Impex International, Inc. The sulfono-γ-AApeptide building block synthesized on solid support Rink-amide resin as previously reported[31,32]. Full details of the chemical synthesis and purification are given in the Supplementary Methods.

**NMR spectroscopy**. The NMR spectra were obtained on a Varian VNMRS 600 MHz spectrometer equipped with four RF channels and a Z-axis-pulse-field gradient cold probe. Sample **8** was measured at a concentration of 4.0 mM in 500 μL CD$_3$OH in a 5 mm NMR tube. The $^1$H shift assignment was achieved by sequential assignment procedures based on DQFCOSY, COSY, zTOCSY, and NOESY measurement. TOCSY and NOESY spectra were acquired with the Wet solvent suppression. All experiments were performed by collecting 4096 points in f2 and 512 points in f1. A DIPSI2 spin lock sequence with a spin lock field of 6k Hz and mixing time of 80 ms were used in zTOCSY. NOESY experiment was carried out using a mix time of 300 ms. Further details of $^1$H shift assignment are given in the Supplementary Methods.

**Circular dichroism**. CD spectra were measured on an Aviv 215 CD spectrometer using a 1 mm path length quartz cuvettes, and compound solutions in methanol were prepared using dry weight of the lyophilized solid followed by dilution to give

the desired concentrations and solvent combination. 10 scans were averaged for each sample, and three times of independent experiments were carried out and the spectra were averaged. The final spectra were normalized by subtracting the average blank spectra. Molar ellipticity [θ] (deg cm$^2$ dmol$^{-1}$) was calculated using the equation:

$$[\theta] = \theta_{obs}/(n \times l \times c \times 10)$$ where $\theta_{obs}$ is the measured ellipticity in millidegrees, while $n$ is the number of side groups, $l$ is path length in centimeter (0.1 cm), and $c$ is the concentration of the α/sulfono-γ-AA peptide in molar units.

**X-ray crystallography**. Lyophilized powders of oligomers **6** (3 mg), **9** (3 mg), and **10** (2 mg) were dissolved in 2 mL of dichloromethane/acetonitrile (20:80, v/v) and then left for slow evaporation at room temperature within 2 days to give crystals. Lyophilized powders of oligomer **4** (2 mg) were dissolved in THF (2 mL) and then pentane (1 mL) was diffused slowly into THF layer, crystals were formed in a week. Crystals of **7** were obtained from slow evaporation of 3 mg/mL solution in chloroform. Oligomer **2** was also crystalized from slow diffusion of pentane into THF in 10 days, however, the crystals were not of good quality for X-ray diffraction (diffraction up to 5.00 Å of resolution only). Foldamers **8** and **9** in 1:1 ratio (4 mg, the racemate **11**) was crystallized from CH$_2$Cl$_2$/CH$_3$CN (60:40, v/v) using slow vaporization over 2 days.

The X-ray diffraction data for all compounds were measured on Bruker D8 Venture PHOTON 100 CMOS system equipped with a Cu K$_\alpha$ INCOATEC Imus micro-focus source (λ = 1.54178 Å). Indexing was performed using APEX3[58] (Difference Vectors method). Data integration and reduction were performed using SaintPlus 6.01[59]. Absorption correction was performed by multi-scan method implemented in SADABS[60]. Space groups were determined using XPREP implemented in APEX3. Structures were solved using SHELXT or SHELXD and refined using SHELXL-2014[61–63] (full-matrix least-squares on $F^2$) through OLEX2 interface program[64]. Two models of crystal structure of oligomer **7** are provided: solvent included (Supplementary Table S4) and solvent masked where disordered solvent molecules and counterions were treated as diffuse using Platon Squeeze procedure (Supplementary Table S5)[65]. Details of crystallization, data collection and refinement are given in the Supplementary Methods.

## Data availability
The authors declare that the data supporting the findings of this study are available within the paper and its Supplementary Information files. The X-ray crystal structures of oligomer **4** (Supplementary Data 1), **6** (Supplementary Data 2), **7** (Supplementary Data 3, Supplementary Data 4), **9** (Supplementary Data 5), **10** (Supplementary Data 6), and **11** (Supplementary Data 7) (the crystal pictures of all the compounds are shown in Supplementary Table S9) have been deposited to the Cambridge Crystallographic Data Centre (CCDC), under deposition numbers CCDC 1541638, 1541639, 1541640, 1976022, 1976024, and 1976034 respectively. These data can be obtained free of charge from The Cambridge Crystallographic Data Centre. and are available as Supplementary Data. Any other datasets are available from the corresponding author on reasonable request.

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

## Acknowledgements

This work was generously supported by NSF 170850 (J.C.), NIH 9R01AI152416-06 (J.C.), and NIH 1R01AG056569-01 (J.C.).

## Author contributions

P.T., M.-M.Z., and J.C. conceived and directed the project and wrote the manuscript. P.T., M.-M.Z., S.X., and W.J. performed the synthesis, purification and characterization. P.T. and M.-M.Z. obtained the crystals. D.C.C. run the 2D NMR experiments and resolved 2D NMR spectra. Y.S. and M.Z. carried out the CD experiments. L.W. collected crystal data and solved the crystal structures. L.-J.M. and Y.H. corrected the manuscript.

## Competing interests

The authors declare no competing interests.
