## [Peer Review File · Communications Chemistry]

Reviewers' comments:

Reviewer #1 (Remarks to the Author):

In this manuscript, the authors elucidated the folded structures of hybrid peptides with sulfono-gamma-AA residues.

Although similar crystal structures of longer oligomers were previously reported, structure analysis in solid and in solution was performed thoroughly. Furthermore, the crystal structure of racemic mixtures shows close contacts between heterochiral peptides. The reviewer therefore thinks this manuscript is worth to be published in Communications Chemistry, after consideration and revision on below points:

Figure 2(d) is missing.

The interaction between foldamers and solvent molecules should be shown in more detail in Figure 3.

The authors compared the backbone torsion angles of foldamers with those of homogeneous L-sulfono-gamma-AA peptides, and mentioned that three backbone torsion angles are different. Please provide the reason for it more clearly. I could not understand why D-sulfono-gamma-AA peptides are used for comparison.

Figure 4b-e: the unit of x axis (λ) should be nm.

Figure 4e: Isodichroic points are observed at around 218 nm and 235 nm. Does it mean a two-state equilibrium?

Reviewer #2 (Remarks to the Author):

This manuscript by Cai and co-workers describes the secondary structure of claimed 413-helices consisting of either D- and or L-amino acid derivatives, as well as an example of interesting supramolecular racemic foldamer complex. The same group has reported an example of secondary structure and a dimer structure in solid state (Ref 31). Foldamers that bear halogen units were also reported (Ref 35) to emphasize the orthogonal halogen interactions in the crystal lattice. However, the general folding propensity in this class of foldamer, as well as solution structures, have not been systematically discussed. In this article, they first reported the structures (both crystal structures and solution structures (CD, 2D-NMR)) of several right-handed helices with different lengths, which demonstrates the generalizability and robustness of folding propensity in this class of foldamer. It is very interesting to see that a foldamer with as short as four sulfono- γ -AA units and five α -amino acid can form stable helical structure, rather than only seven and more sulfono- γ -AA units were used in the previous report. Another significant finding in this article is that it demonstrates the ease to control the helical handedness in this class of foldamer. One can often predict the handedness of a peptidic foldamer by looking at the chiral configuration of the amino acid in the backbone, but it is still tricky especially when the hybrid foldamers are constituted of both natural and unnatural amino acid backbone. The authors show two left-handed helices using the mirror configured amino acids, by using both X-ray single crystallography and CD, which add more novelty into the foldamer structures. Furthermore, one more novel result is that the unnatural peptides-derived heterochiral association in the manuscript formed an interesting packing model akin to two gears of opposite sense. It is pleasant to read this manuscript and the reviewer agrees that the current work warrants to publish in Communications Chemistry after revisions. The above are the review I made for this

manuscript previously submitted for another journal. As the authors have addressed my previous concerns, I recommend the work for Communications Chemistry after a minor revision: I suggest the authors to add the crystal packing of left-handed helices to SI, in Fig S3.

Reviewer #3 (Remarks to the Author):

Cai et al wrote a complete structural study about the folding propensity of α /sulfonyl-g-AA peptidic foldamers. The sequences of the present manuscript are directly linked to a paper published by the same authors in JACS (2018) in which they described among other things a similar artificial peptide (to compound 8) and its crystal structure. The present study seems to me a follow-up paper (structural) using different technique such as NMR, CD and more particularly X-ray diffraction.

The folding propensity of these objects is well highlighted in the solid state and in solution. However relying strongly on x-ray structures the authors could have paid more attention to the refinement of their crystal structures. All structures need to be checked again and some cif files upgraded (Z values are wrong for structures 89, 91, 92 and 93). The raw data for structure 90 are of poor quality, the authors refine the structure in the monoclinic space group P2(1) although the cell parameters seem to be very close to a tetragonal system (a axis close to b axis and beta angle of 90°). Solving and refining the structure in a tetragonal (or even orthorhombic) space group needs to be considered and commented if not possible. Finally, if the structure is refined in the monoclinic space group treatment of the disordered solvent needs to be taken into account. On the P2(1) structure provided, the R_{cryst} factor drops to 8% if you remove the very disordered solvent molecules and flatten the electron density using the Squeeze procedure.

To conclude, I would like to point out that the most innovative part is at the very end of the manuscript and concern the hetero-chiral coiled coil foldamer. The authors just scratched the surface of thing on this phenomenon. There is a clear lack of solution studies (CD, NMR ...) to assess the stability of the coiled other than in the solid state. More references on artificial coiled coil would be welcome.

Overall I recommend the publication after major modifications and additions.

Reviewers' comments:

Reviewer #1 (Remarks to the Author):

In this manuscript, the authors elucidated the folded structures of hybrid peptides with sulfonogamma- AA residues.

Although similar crystal structures of longer oligomers were previously reported, structure analysis in solid and in solution was performed thoroughly. Furthermore, the crystal structure of racemic mixtures shows close contacts between heterochiral peptides. The reviewer therefore thinks this manuscript is worth to be published in Communications Chemistry, after consideration and revision on below points:

Thank reviewer for the good comments.

Figure 2(d) is missing.

Thank reviewer for your carefully reading. Figure 2d has been added in the revised manuscript. Please refer to the change on Page 6.

The interaction between foldamers and solvent molecules should be shown in more detail in Figure 3.

Thank reviewer for the suggestion. We have made some revision on this comment. The interaction between foldamers and heavily disordered solvent molecules (CH_3CN , CHCl_3 , or THF/pentane) are so weak than other weak interactions including $\text{C}-\text{Cl}\cdots\text{O}=\text{C}$ (3.2 Å, in 11-mer **4**), $\text{C}-\text{Cl}\cdots\pi$ (3.2 Å, in 11-mer **6**), $\text{C}-\text{Cl}\cdots\text{Cl}-\text{C}$ (3.4 Å, in 11-mer **6** and 14-mer **7**) and other Van der Waals interactions, so it won't affect the assembly/packing of the foldamers. Therefore, the disordered solvent THFs/pentanes (in 11-mer **4**), acetonitriles (in 13-mer **6**) or chloroforms (in 14-mer **7**) are excluded from the crystal lattice in Figure **3**. This has been added in the caption of Figure **3** in the revised manuscript.

The authors compared the backbone torsion angles of foldamers with those of homogeneous L-sulfono- γ -AA peptides, and mentioned that three backbone torsion angles are different. Please provide the reason for it more clearly. I could not understand why D-sulfono- γ -AA peptides are used for comparison.

Thank reviewer for the suggestion. We have revised this part in the revised manuscript, please refer to the highlights on Page 11. Anyway, the result is still beyond our understanding at this point, since the homogeneous L-sulfono- γ -AA foldamers form unexpected left-handed helices, rather than the right-handed helices in this study. In addition, we have removed the discussion of comparison with D-sulfono- γ -AA peptides since it does not make sense to compare with them on the backbone torsion angles.

Figure 4b-e: the unit of x axis (λ) should be nm.

Thank reviewer for your carefully reading. The unit of x axis was labeled as “nm” in the revised manuscript now, the unit of x axis in Figure 5a was also revised.

Figure 4e: Isodichroic points are observed at around 218 nm and 235 nm. Does it mean a two-state equilibrium?

Thank reviewer for the great suggestion. We believe it indicates a two-state equilibrium. In the revised manuscript on Page 15, we have added more discussion, “the curves share isodichroic points at 218 nm and 237 nm (Figure 4e), suggesting transitions between two structural states (unfolded and folded) with few intermediate species, the two-state equilibrium also indicated the possible presence of aggregation of the foldamers in solution”.

Reviewer #2 (Remarks to the Author):

This manuscript by Cai and co-workers describes the secondary structure of claimed 413-helices consisting of either D- and or L-amino acid derivatives, as well as an example of interesting supramolecular racemic foldamer complex. The same group has reported an example of secondary structure and a dimer structure in solid state (Ref 31). Foldamers that bear halogen units were also reported (Ref 35) to emphasize the orthogonal halogen interactions in the crystal lattice. However, the general folding propensity in this class of foldamer, as well as solution structures, have not been systematically discussed.

In this article, they first reported the structures (both crystal structures and solution structures (CD, 2D-NMR)) of several right-handed helices with different lengths, which demonstrates the generalizability and robustness of folding propensity in this class of foldamer. It is very interesting to see that a foldamer with as short as four sulfono- γ -AA units and five α -amino acid can form stable helical structure, rather than only seven and more sulfono- γ -AA units were used in the previous report.

Another significant finding in this article is that it demonstrates the ease to control the helical handedness in this class of foldamer. One can often predict the handedness of a peptidic foldamer by looking at the chiral configuration of the amino acid in the backbone, but it is still tricky especially when the hybrid foldamers are constituted of both natural and unnatural amino acid backbone. The authors show two left-handed helices using the mirror configured amino acids, by using both X-ray single crystallography and CD, which add more novelty into the foldamer structures. Furthermore, one more novel result is that the unnatural peptides-derived heterochiral association in the manuscript formed an interesting packing model akin to two gears of opposite sense. It is pleasant to read this manuscript and the reviewer agrees that the current work warrants to publish in Communications Chemistry after revisions. The above are the review I made for this manuscript previously submitted for another journal. As the authors have addressed my previous concerns, I recommend the work for Communications Chemistry after a minor revision: I suggest the authors to add the crystal packing of left-handed helices to SI, in Fig S3.

Thank reviewer for the comments. As you mentioned, we have received comments previously and made revision according to your previous comments. In the revised manuscript, we have added the crystal packing of left-handed helices into the Supporting Information, please refer to Page S11.

Reviewer #3 (Remarks to the Author):

Cai et al wrote a complete structural study about the folding propensity of α /sulfonyl-g-AA peptidic foldamers. The sequences of the present manuscript are directly linked to a paper published by the same authors in JACS (2018) in which they described among other things a similar artificial peptide (to compound 8) and its crystal structure. The present study seems to me a follow-up paper (structural) using different technique such as NMR, CD and more particularly X-ray diffraction.

Thank reviewer for the suggestion. We politely disagree with you. As other reviewers pointed out, the general folding propensity in this class of foldamer, as well as solution structures, have not been systematically discussed so far but were studied in this manuscript. We also demonstrate the robust control of the helical handedness in this class of foldamer for the first time. In addition, it is not always easy to predict the handedness of a peptidic foldamer based on the chiral configuration of the backbone amino acid, especially when the foldamers are constituted of hybrid amino acid backbone, as also pointed by another reviewer. The authors show two left-handed helices using the mirror configured amino acids, it seems easy to predict and understand now, but one still need to be cautious to fully understand the secondary structures of the left-handed helices at certain degree.

The folding propensity of these objects is well highlighted in the solid state and in solution. However relying strongly on x-ray structures the authors could have paid more attention to the refinement of their crystal structures. All structures need to be checked again and some cif files upgraded (Z values are wrong for structures 89, 91, 92 and 93).

Thank reviewer for the suggestion. We have conducted more refinements on the crystallography data and addressed the concerns you had before. They have been revised by the crystallographic expert Dr. Lukasz Wojtas, who is coauthor of the manuscript. Full CIF file (including ins/hkl) has

now been processed with newer version of CheckCIF clearing the nomenclature errors. An updated checkCIF has been submitted.

Specifically, Z values of Oligomer **6** and Oligomer **10** were changed to reflect the exact number of polypeptide in unit cell. The number is not a positive integer due to the specific type of disorder present that cannot be modeled with classical crystallography and single unit cell - please see the supporting information for discussion. We cannot see Z-value issues with remaining structures where Z value is chosen to reflect the number of polypeptide molecules in the unit cell. Chosen Z-values are not consistent with Platon output due to inability of Platon to detect isolated molecules in pseudo-polymeric model.

The raw data for structure 90 are of poor quality, the authors refine the structure in the monoclinic space group P2(1) although the cell parameters seem to be very close to a tetragonal system (a axis close to b axis and beta angle of 90°). Solving and refining the structure in a tetragonal (or even orthorhombic) space group needs to be considered and commented if not possible. Finally, if the structure is refined in the monoclinic space group treatment of the disordered solvent needs to be taken into account. On the P2(1) structure provided, the R_{cryt} factor drops to 8% if you remove the very disordered solvent molecules and flatten the electron density using the Squeeze procedure.

The diffraction data for oligomer **7** (we assume this is the compound the reviewer was referring as structure 90) is of lower resolution, crystal was twinned and contains disordered solvent in structural voids. Those factors contribute to lower quality of structural model. The data quality however is good enough to resolve secondary structure and model some of solvent molecules. The model was refined with restraints.

It is impossible for this structure to have higher symmetry because there is only one molecule in ASU and the molecule itself is asymmetric. Due to the specific packing the lattice is pseudo-tetragonal and crystal was refined as pseudo-merohedral twin.

Typically, solvent mask lowers R-factor but we believe solvent should be modeled whenever it is possible as it is important part of crystal structure. To satisfy reviewer we now report solvent masked model of the structure, but we still insist on publication of revised model with solvent present. Two CIF files are now provided as part of supporting information for this structure.

To conclude, I would like to point out that the most innovative part is at the very end of the manuscript and concern the hetero-chiral coiled coil foldamer. The authors just scratched the surface of thing on this phenomenon. There is a clear lack of solution studies (CD, NMR ...) to assess the stability of the coiled other than in the solid state. More references on artificial coiled coil would be welcome.

Thank reviewer for the good suggestion. We agree with the reviewer that the most innovative part is at the very end of the manuscript regarding the hetero-chiral coiled coil foldamer, however, we politely disagree with you that this affects the general novelty of the current manuscript. As we mentioned in response to your first comment, and as other reviewers pointed out, we have addressed the general folding propensity in this class of foldamer, as well as solution structures and the handedness control of a peptidic foldamer based on the chiral configuration of the backbone amino acid, which have not been systematically discussed so far. In addition, we brought up with the

hetero-chiral coiled coil foldamer on this phenomenon as you commented, but the solution structures of such a structure will be different with a single-handedness peptidic foldamer, and are still under investigation, hopefully we can assess the stability of the coiled in solutions unambiguously and report the result in the community soon.

As far as the references, we have added more references for heterochiral coiled coil in the revised manuscript (ref. 54, 55, 56, 37, and 38).

Overall I recommend the publication after major modifications and additions.

Thank reviewer for the comment. As we have addressed the comments on the crystallography data and refinements, we wish you think the revised version is acceptable for publication in *Communications Chemistry*.

Reviewers' comments:

Reviewer #1 (Remarks to the Author):

The authors responded and revised to each of the reviewer's comments and suggestions well. This reviewer is satisfied with that. The manuscript is worth to be published in Communications Chemistry.

Reviewer #3 (Remarks to the Author):

none

3/4/2021

Dear Reviewer 3#,

We are very thankful for your insightful and constructive comments. We have double checked all the structures especially oligomers **6** and **10** where the Zs are not integer. According to your recent comments, we added more detailed discussion as a response. However, our refinement strategy is based on strategies employed by protein crystallographers where electron density difference maps (Fo-Fc and 2Fo-Fc) are used to identify solvent molecules partly based on shape of continuous electron density, hence the strategies employed for refinements are different from that for the small molecules. In our opinion, your recent comments are a little too broad and without specific details or suggestions. We have tried our best to answer them. We hope you are satisfied with our response and our manuscript can be accepted for publication.

Thank you very much for your time!

Best regards,

Jianfeng Cai, Ph.D.
USF Preeminent Professor
Department of Chemistry
University of South Florida
Tampa, FL 33620
E-mail: jianfengcai@usf.edu

Reviewer comments:

- concerning the resolution of the X-ray structures, one can observe issues with the way they deal with the partial occupancies of the disordered molecules and the determination of Z and Z'.

Thank reviewer for the suggestion. Although the values of Z/Z' are arbitrary to some extent it is common practice to choose the value in order to obtain correct formula in CIF file especially in case of discrete molecules.

In the asymmetric unit there is fraction of molecule in each case 1/3.5 (for oligomer with ratio 6:7, oligomer **6**), 1/4 (for oligomer with ratio 7:8, oligomer **8**), 1/4.5 (for oligomer with ratio 8:9, oligomer **10**). This is NOT USUAL because reported oligomers do not have any internal symmetry. The reason for this is specific disorder described in SI and resulting in smaller than expected unit cell parameter in the direction of oligomer's axis (Directions a,b~17.1Å). The translational disorder of hydrogen bonded columns of chains in the structure causes the structure to appear as made of infinite chains. In reality there are small gaps where hydrogen bonds are present, small fraction of

gamma peptide is missing and occupancy is set to lower value to reflect that and to obtain correct ratio of peptides in molecule.

The chosen Z value reflects the exact number of molecules in the unit cell (which does not have to be integer as super cell can be found that contains integer number of molecules) and returns proper chemical formula of oligomers.

Regarding Z and Z': Z and Z' are not independent - indeed $Z'=Z/N$ (N is space group multiplicity and is equal to 8 for P4₃2₁2 space group). In order to calculate unit cell content, we have to multiply asymmetric unit content by space group multiplicity. We obtain 2.286, 2 and 1.778 molecules in one unit cell correspondingly. What follows is that because of specific disorder the number of molecules in unit cell the structure is modeled in is not an integer number. No other indexing results are possible with this data and there is no possibility for mis-indexing.

Compound	Formula	Ratio	Z	Z'=Z/8	Unit cell, Tetragonal P4 ₃ 2 ₁ 2	Volume [Å ³]
6/ pt_g_36_2_0m	C ₈₉ H ₁₁₈ Cl ₆ N ₂₀ O ₂₆ S ₆	6:7	2 ² / ₇ =2.285	2/7=0.286	17.1537(4), 28.9825(8)	8528.1(5)
9/ JC_TP_H_108_1	C ₁₀₃ H ₁₃₆ Cl ₇ N ₂₃ O ₃₀ S ₇	7:8	16/8=2	¼=0.25	17.1371(4), 29.0104(8)	8519.8(5)
10/ TP_H_108_3	C ₁₁₇ H ₁₅₄ Cl ₈ N ₂₆ O ₃₄ S ₈	8:9	1 ⁷ / ₉ =1.778	2/9=0.222	17.1367(3), 28.985(1)	8511.9(4)

- There is nothing dramatic for the discussion of the structures themselves yet the cif is not fully correct.

Thank reviewer for the suggestion. All structures contain large structural voids with heavily disordered solvent. Our refinement strategy is based on strategies employed by protein crystallographers where electron density difference maps (Fo-Fc and 2Fo-Fc) are used to identify solvent molecules based in part on shape of continuous electron density. We have located all solvent molecules using WinCoot program and refined them with restraints. We always attempt to refine occupancies to determine approximate values for structures with diffuse (but still conclusive) electron density peaks and then fix them while taking into account presence of surrounding molecules. Benefits of including solvent molecules when possible, is obvious from the point of view of possible future computational research where solvent is important part of modelling. Therefore, we chose it over “Squeeze” approach where effectively part of the structure is removed. Squeeze method can be used at any time after publication as reflection data is provided, while solvent modelling is time consuming and cannot be done by non-expert.

REVIEWERS' COMMENTS:

Reviewer #3 (Remarks to the Author):

[Editorial note: this reviewer provided no further comments to the authors.]